# A Single-Joint Worm-like Robot Inspired by Geomagnetic Navigation

**Dong Mei** , **Xin Zhao** , **Gangqiang Tang** , **Jianfeng Wang, Chun Zhao, Chunxu Li and Yanjie Wang** *

Jiangsu Provincial Key Laboratory of Special Robot Technology, Hohai University, Changzhou Campus, Changzhou 213022, China
* Correspondence: yj.wang1985@gmail.com

**Abstract:** Inspired by identifying directions through the geomagnetic field for migrating birds, in this work, we proposed and fabricated a single-joint worm-like robot with a centimeter scale, the motion of which could be easily guided by a magnet. The robot consists of a pneumatic deformable bellow and a permanent magnet fixed in the bellow's head that will generate magnetic force and friction. Firstly, in order to clarify the actuating mechanism, we derived the relationship between the elongation of the bellows and the air pressure through the Yeoh constitutive model, which was utilized to optimize the structural parameters of the bellow. Then the casting method is introduced to fabricate the silicone bellow with a size of 20 mm in diameter and 28 mm in length. The manufacturing error of the bellow was evaluated by 3D laser scanning technology. Thereafter, the robot's moving posture was analyzed by considering the force and corresponding motion state, and the analysis model was established by mechanics theory. The experimental results show that the worm-like robot's maximum speed can reach 9.6 mm/s on the cardboard. Meanwhile, it exhibits excellent environmental adaptability that can move in pipelines with a diameter of 21 mm, 32 mm, 40 mm, and 50 mm, and surfaces with different roughness. Moreover, the robot's motion was successfully guided under the presence of the magnetic field, which shows great potential for pipeline detection applications.

**Keywords:** worm-like robot; magnetic navigation; pneumatic actuator; structural design; fabrication; characterization; piping application

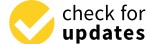



## 1. Introduction

Nature provides a steady stream of inspiration for engineers to solve problems [1]. For a long time, the principle of bionic motion has been widely applied in the design of various worm-like robots and has attracted worldwide research attention [2]. Compared with the traditional wheel-legged bionic robotics, the worm-like robot, which generates motion by imitating simple changes in the body shape of worms (earthworms and inchworms), can be ideal for navigating long narrow spaces due to its compact structure, lightweight, and small movement space, and has been widely used in pipeline maintenance, medical endoscopy, and other fields [3]. The most previously developed peristaltic robots are built with rigid materials and structures [4–6]. Usually, these rigid robots have poor environmental adaptability due to the "hard" nature of their characteristics, which can only crawl in a specific pipeline environment and possibly damage the pipeline's surface due to improper operation. This significantly limits the peristaltic robot's capability to be used in environments with different pipeline sizes. With the development of materials, biomechanics, and manufacturing technologies, soft robotic technology has attracted the extensive attention of researchers. Compared with rigid robots, soft robots, which have continuous deformation, infinite degrees of freedom theoretically, more robust environmental adaptability, and safely interact with humans [7,8], are made of flexible or soft materials (a Young's modulus of $10^4$–$10^9$ Pa) and can arbitrarily change their shape to pass the complex constrained environments. The emergence of soft robots provides considerable prospects for peristaltic robots.

Most of the existing soft worm-like robots are designed in multiple-segmented structures, which can realize the effective contraction-expansion motion like earthworms or the $\Omega$ motion like inchworms by actuating this segmented structure in a predesigned sequence [9]. For instance, Onal et al. [10] presented a worm-like origami robot driven by shape memory alloy (SMA), which can realize the crawling motion by controlling the elongation and contraction through different SMA. Pfeil et al. [11] proposed a bellows structured robot, which was actuated by dielectric elastomers (DE) and can stably realize the peristalsis on the horizontal surface. However, the efficiency of the worm-like robot's locomotion actuated by the SMA spring or DE did not exhibit satisfactory performance as a result of the intractability of controlling the SMA and DE in the motion process. In contrast, pneumatically is widely adopted for driving soft worm-like robots because of its low-cost design and easy control. Furthermore, pneumatic actuation has high driving efficiency and achieved many impressive achievements [12–14]. The standard structure of such worm-like robots has three parts: two chambers for anchoring and one drive actuator for locomotion. For example, Verma et al. [13] designed a peristaltic wall-climbing robot with three chambers, two of which are used as radial actuators, while the third is used as a liner actuator. The robot achieves bidirectional motion along pipelines by periodically changing the input air pressure in each chamber. A similar worm-like robot was also proposed by Liu et al. [14], and the vertical load capacity could reach 1 kg. However, such robots only have one degree of freedom (DOF), which cannot adjust the heading directions flexibly and actively. In order to achieve more flexible motion capabilities, such as two-dimensional steering motion, the design of the robot's structure and control system must be more complex. For example, in [14], Liu et al. superimposed multiple different pneumatic modules on the basis of the robot's peristaltic module, which realized the robot's steering in the pipeline. Zhang et al. [15] presented a multi-chamber actuator as a liner actuator that can realize the turning of the robot by inflating different chambers to bend the linear actuator. On the one hand, these complex structures undoubtedly increase the difficulty of manufacturing and controlling the robot. On the other hand, the aforementioned peristaltic robot can only crawl inside the pipeline, and its plane crawling ability has not been verified. Moreover, the diameter of the crawling pipelines is limited by the deformation range of its expansion segments. The expansion can cause the pipeline to be blocked, which dramatically limits the worm robot's application. Although, some researchers have conducted the creeping on different environments, such as pipelines and planes, based on the principle of friction anisotropy. An impressive example is a worm robot proposed by Ge et al. [16] that can realize the forward or backward movement in the pipeline and planes by changing the friction coefficient of the two end segments, but how to realize the steering is still a complex problem.

In view of the above problems, we designed a single-joint centimeter-scale magnetic navigation worm-like robot in this study inspired by the use of geomagnetic field navigation in the migration of migratory birds. As shown in Figure 1a, the migratory birds modify their flight routes according to the change of geomagnetic field during the migration process to ensure the correct migration direction. As shown in Figure 1b, the designed robot is composed of a bellows actuator and a permanent navigation magnet, which provide power and directional traction, respectively. The forward movement can be realized by the principle of friction anisotropy and the force-bending characteristics of the bellows based on the principle of inchworm peristalsis, while the robot's steering and motion path can be guided by adjusting the magnetic force direction.

In general, the main highlights of this work include the following:

(1) Based on the principle of worm motion, a single-joint centimeter-scale magnetic navigation worm-like robot that can crawl on different diameters of pipelines and surfaces with different roughness without being limited by the range of radial segment deformation is designed using the force-bending characteristics of bellows and the principle of friction anisotropy in this study.

(2) Inspired by utilizing the geomagnetic field for migrating birds to identify directions, the robot can be turned and guided by changing the direction of the magnetic field, which simplifies the robot's structural design and reduces the difficulty of manufacturing and control.

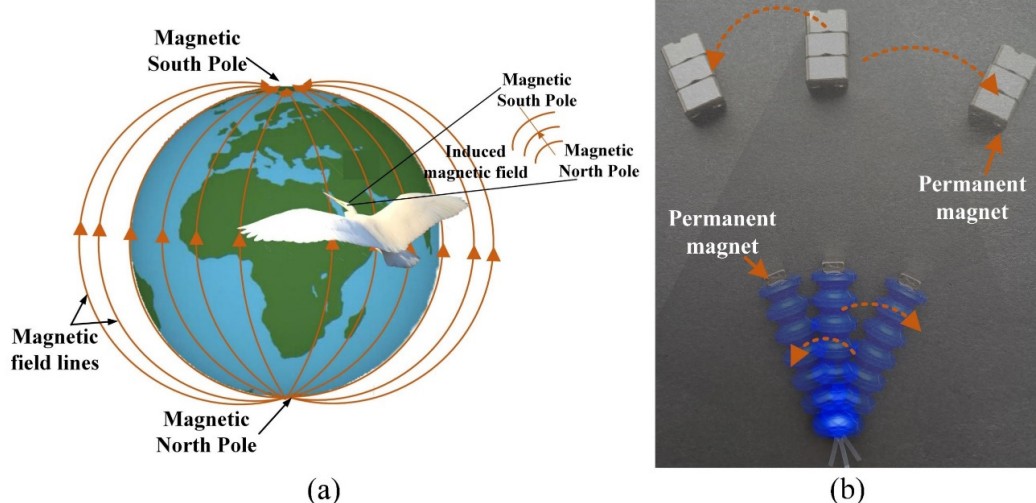

(a)　　　　　　　　　　　　　　　　　　　　　　　(b)

**Figure 1.** The schematic of the magnetic field navigation (**a**) the schematic diagram of navigation of migratory birds; (**b**) the schematic diagram of navigation of the robot.

The remainder of this work is presented as follows. The structure parameters of the pneumatic actuator and the model of air pressure and elongation are established in Section 2. Section 3 analyzes the motion principle of the robot and establishes the mechanical model. The experiments to verify the bellows characteristics, motion characteristics, and navigation characteristics are in Section 4. Finally, the conclusion and future works are concluded in Section 5.

## 2. Design and Fabrication

In this section, the structure and fabrication process of the robot are described, and the mathematical model between the air pressure and the elongation of the bellows actuator is established based on the Yeoh constitutive model. Then, the Abaqus is used to analyze the influence of the rectangular chamber number and wall thickness on the performance of bellows with the same conditions, and the actuator structure is optimized by its results. Finally, the robot actuator is manufactured according to the determined structural parameters.

### 2.1. Design Concept and Structure of the Robot

This work is inspired by migratory birds, which can adjust their flight direction according to the direction of the induced geomagnetic field. As shown in Figure 1b, the robot's structure is mainly composed of a motion module and a navigation module. The motion module is made of flexible silicone rubber that adopts a bellows actuator structure to power it. The navigation module is a rigid permanent magnet pasted on the robot head that guides the robot and furnishes a diffident friction coefficient.

As the unique power unit, the bellows actuator's performance directly determines the crawling efficiency of the robot. As shown in Figure 2, the rectangular bellows structure is adopted in this paper to realize the expansion and contraction movement of the robot by inflating and deflating. The dimensions of the actuator in the initial state are described by the height $l$, the single chamber height $x$, the chamber radius $r$, the thickness $t$, and the radius $R$. The dimensions of the actuator in the deformed state are described by the height $l_a$, the single chamber height $x_a$, the chamber radius $r_a$, the thickness $t_a$, and the radius $R_a$. The initial height $l$ can be calculated by $l = nx$, where $n$ is the chamber's numbers, the change of height of the actuator between the initial state and deformed state or named

the actuator's elongation is termed as $\Delta l$, and the actuator's expansion is calculated as $w = 2(R_a - R)$.

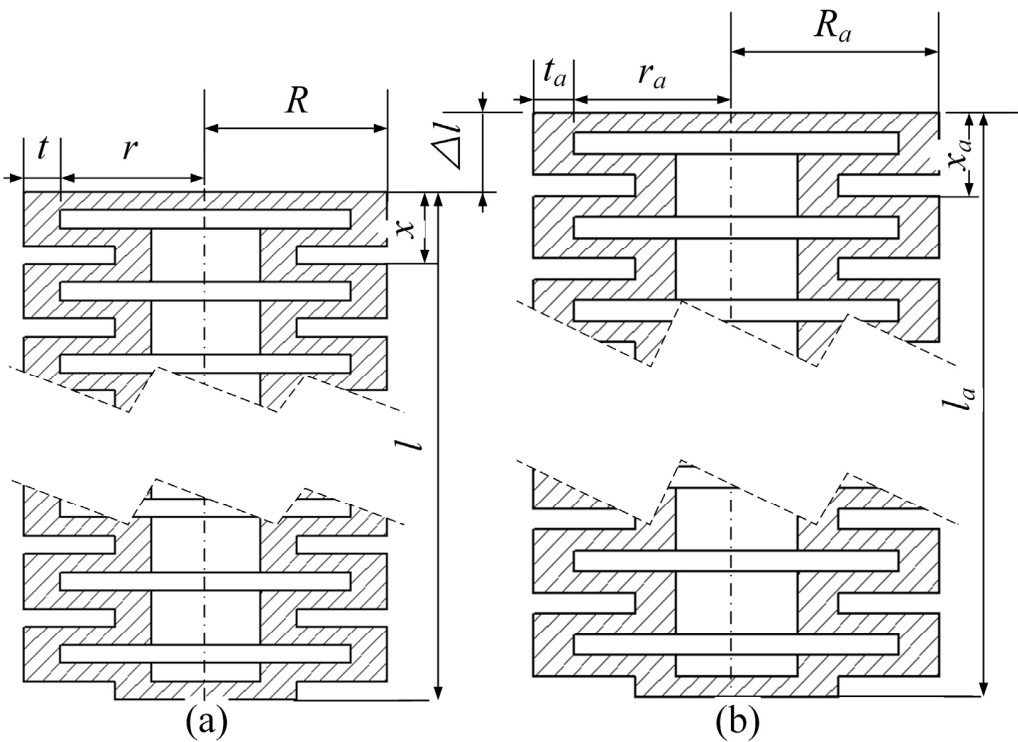

**Figure 2.** Cross section broken view of the bellows actuator in the initial state and deformed state (**a**) Initial state; (**b**) Deformation state.

Since silicone rubber is a kind of nonlinear material with hyperelasticity, large deformation, and incompressibility, we use the Yeoh model to analyze the relationship between pressure and elongation of the actuator. The Yeoh model strain energy function is expressed as follows:

$$W = W(I_1, I_2, I_3) \tag{1}$$

$$\begin{cases} I_1 = \lambda_1{}^2 + \lambda_2{}^2 + \lambda_3{}^2 \\ I_2 = \lambda_1{}^2\lambda_2{}^2 + \lambda_2{}^2\lambda_3{}^2 + \lambda_1{}^2\lambda_3{}^2 \\ I_3 = \lambda_1{}^2\lambda_2{}^2\lambda_3{}^2 \end{cases} \tag{2}$$

where $I_1$, $I_2$, $I_3$ are strain invariants, and $\lambda_1$, $\lambda_2$, $\lambda_3$ represent the main stretch ratios in three directions and it can be written as:

$$\begin{cases} \lambda_1 = \frac{l_a}{l} = \frac{nx_a}{nx} \\ \lambda_2 = \frac{r_a}{r} = \frac{R_a - t_a}{R - t} \\ \lambda_3 = t_a/t \end{cases} \tag{3}$$

Due to the incompressible nature of the silicone rubber materials, the strain invariant $I_3$ can be rewritten as:

$$I_3 = \lambda_1{}^2\lambda_2{}^2\lambda_3{}^2 = 1 \tag{4}$$

Assuming that the bellows base does not deform in the width direction, that is $\lambda_3 = t_a/t = 1$. Simultaneous Equations (2) and (3), it can be obtained that:

$$\begin{cases} \lambda_2 = \frac{1}{\lambda_1} \\ I_1 = I_2 = \lambda_1{}^2 + \frac{1}{\lambda_1{}^2} + 1 \end{cases} \tag{5}$$

According to Equations (1) and (5), the third-order Yeoh model can be written as:

$$W = C_1(I_1 - 3) + C_2(I_2 - 3)^2 + C_3(I_3 - 3)^3 = C_1\left(\frac{l_a^2}{l^2} + \frac{l^2}{l_a^2} - 2\right) + C_2\left(\frac{l_a^2}{l^2} + \frac{l^2}{l_a^2} - 2\right)^2 + 8C_3 \quad (6)$$

where $C_1$, $C_2$, $C_3$ are the coefficients of the material, which can be identified by least square fitting using the experiment data of uniaxial tension tests as $C_1 = 0.12$ Mpa, $C_2 = 0.023$ Mpa, and $C_3 = 0.00006$ Mpa.

According to the law of conservation of energy, the change of the strain energy is equal to the work done by the pressure [17], namely:

$$PdV_a = V_b dW_a \quad (7)$$

where $P$ represents the pressure, $V_a = \pi(r_a - t_a)^2 l_a$ is the volume of the deformed chamber, $W_a$ is the elastic density strain energy, $dV_a$ and $dW_a$ represent, respectively, the variation of $V_a$ and $W_a$, and $V_b$ is the volume of silicone after the actuator deformed. It is worth noting that $V_b$ can be regarded as a constant value due to the incompressibility of the silicone material. It should also be noted that both $V_a$ and $W_a$ are unary functions of $l_a$, and the relationship between pressure $P$ and the elongation $l_a$ can be obtained with the equation above, and it can be written as:

$$P = V_b \frac{dW_a}{dV_a} = f(l_a) \quad (8)$$

The relationship between the bellows actuator elongation and air pressure can also be obtained:

$$\begin{cases} l_a = f^{-1}(p) \\ \Delta l = l_a - l \end{cases} \quad (9)$$

It can be seen from Equation (3) that when the height $x$ of a single rectangular chamber and the radius $R$ of the actuator are constant, the axial elongation ratio of the actuator is related to the number of rectangular chambers $n$, and the radial expansion ratio is related to the bellows thickness $t$. According to Equations (5), (6), and (9), it can be inferred that the number of rectangular chambers and the actuator thickness are transformed into a relationship with the bellows elongation. To explore more intuitively the influence of the rectangular chamber numbers and the actuator thickness on the bellows actuator performance and provides a more accurate basis for the design parameters of the bellows actuator. The bellows actuator's initial parameters are defined as $l = 28$ mm, $R = 10$ mm, $x = 3$ mm, and $t = 1$ mm. Then, the number of the rectangular chamber is made to be $n = 4$, $n = 5$, and $n = 7$, respectively, and the air pressure from 0 kpa to 50 kpa was applied to the bellows actuator for simulation calculation. The results are shown in Figure 3a,b.

As shown in Figure 3a,b, with the increase in the number of rectangular chambers, the bellows actuator elongation is positively correlated with the number of the rectangular chamber under the same conditions. Although the expansion increased with the number of rectangular chambers, there is not much difference between the expansion dimensions of the three candidates under the same thickness. Therefore, we selected the number of the rectangular chamber as $n = 7$, and simulation experiments were carried out with thickness $t = 1$ mm, $t = 2$ mm, and $t = 3$ mm, respectively. The results are shown in Figure 3c,d.

It can be seen from Figure 3c,d that the bellows actuator elongation is negatively correlated with the thickness, and its expansion also decreases with the increase of the thickness of the bellows. Nevertheless, it is worth noting that with the increase in thickness, the expansion of the bellows decreases in multiples. The maximum expansion of $t = 2$ mm is only 44.9% of the maximum expansion of $t = 1$ mm, and the maximum expansion of $t = 3$ mm is 41.8% of $t = 2$ mm. However, the numerical difference between $t = 3$ mm and $t = 2$ mm can be negligible compared with $t = 2$ mm and $t = 1$ mm. In order to avoid the influence of excessive expansion on the robot's motion, which will decrease the bending performance, the number of the rectangular chamber and thickness are selected as $n = 7$

and $t = 2$ mm, respectively, on the basis of ensuring elongation. The structural parameters of the bellows actuator are shown in Table 1.

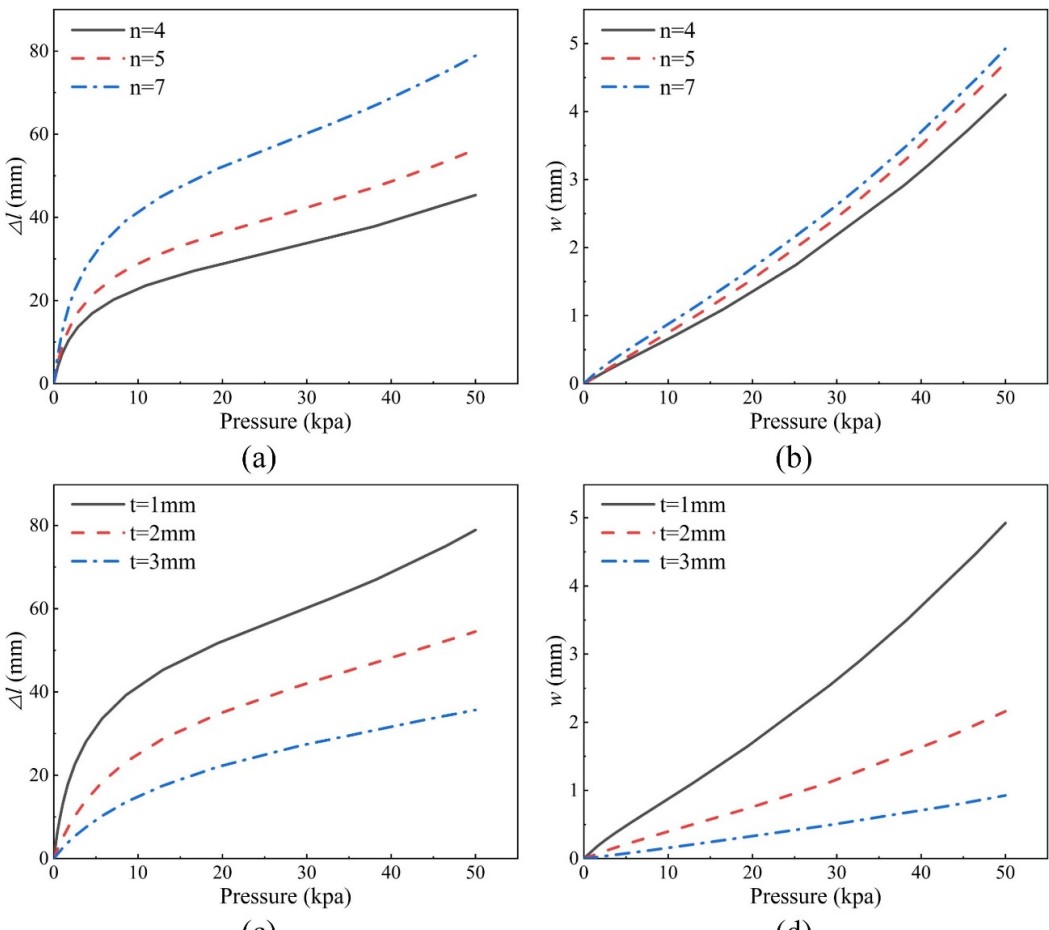

**Figure 3.** The influence of different structural parameters on bellows deformation. (**a**) The effect of the rectangular chamber's numbers on the elongation; (**b**) The effect of the rectangular chamber's numbers on the expansion; (**c**) The effect of the thickness of the bellows on the elongation; (**d**) the effect of the bellows thickness on the expansion.

**Table 1.** The bellows actuator's structure paraments.

| Paraments | Value |
|---|---|
| Height (*l.* mm) | 28 |
| Radius (*r.* mm) | 8 |
| Thickness (*t.* mm) | 2 |
| Numbers (*n*) | 7 |

## 2.2. Fabrication

The main existing manufacturing methods for soft robots are multi-material 3D printing techniques [18], lost wax casting methods [19], and shape deposition manufacturing [20,21]. To further simplify the manufacturing process of the bellows actuator, we adopted the traditional shape deposition method to cast the bellows, and the bodies were connected by gluing. Furthermore, the laser scanner was used to evaluate the error between the actual casting results and the ideal model. The manufacturing process is shown in Figure 4.

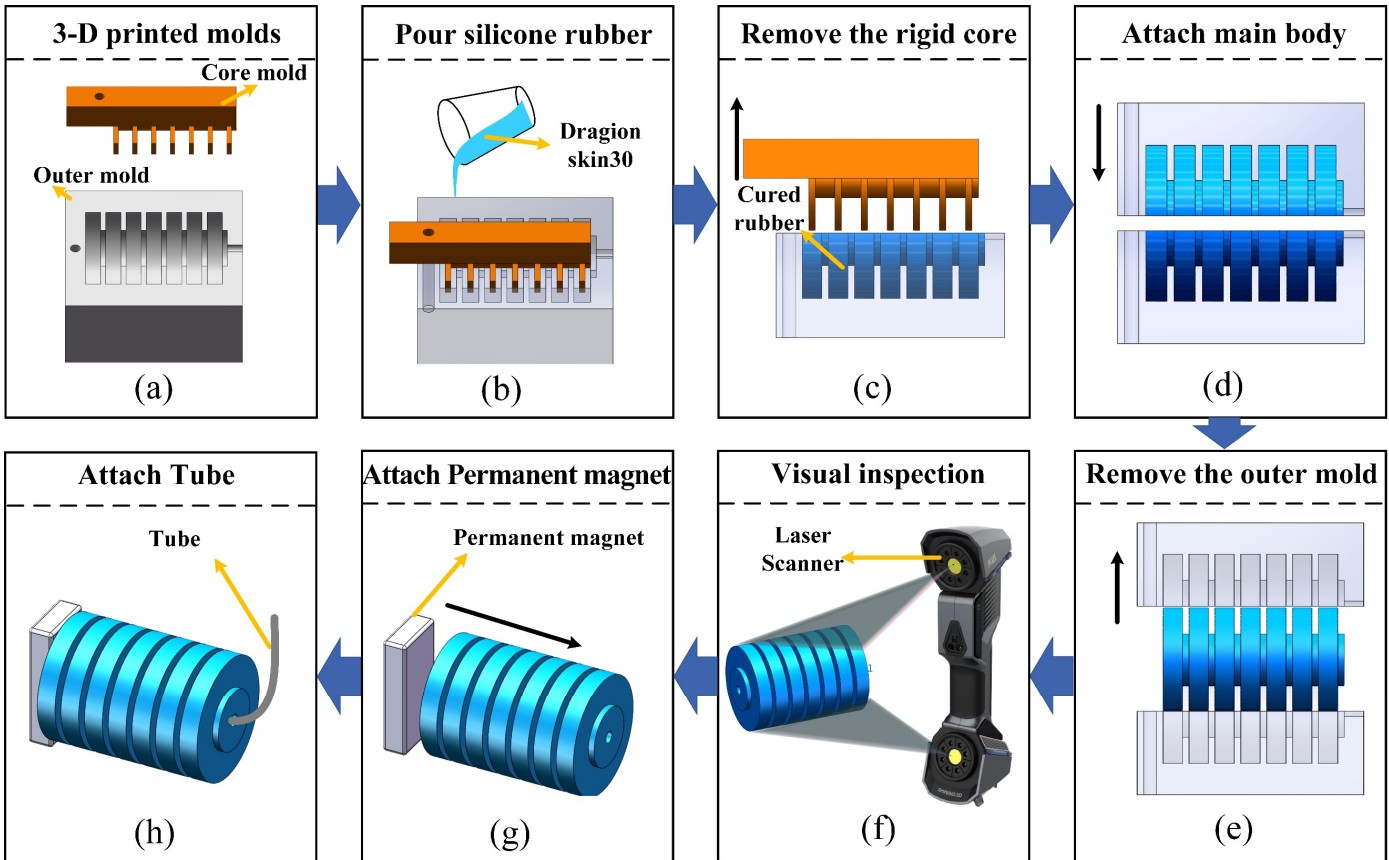

**Figure 4.** The manufacturing process of the robot (**a**) Prepare molds with 3-D printing; (**b**) Pour silicone rubber pre-elastomer (Dragon-skin30 from Smooth-On, Inc, Macungie, PA, USA); (**c**) Remove the rigid core; (**d**) Gluing the two parts; (**e**) Detach the outer molds; (**f**) Detect the soft body by laser scanner; (**g**) Assemble the soft body and a permanent magnet; (**h**) Insert rubber tube and sealing.

First, we 3-D printed molds with the shape of bellows and fixed the rigid core into the outer mold, and then poured silicone rubber pre-elastomer (Dragon-skin30 from Smooth-On, Inc.) into the mold (see Figure 4a,b). As the silicone rubber is cured, we removed the rigid core and fix the solidified silicone rubber into the outer mold (see Figure 4c). Then, the outer edges of the two parts of the actuator were coated with glue, and the outer mold was aligned and pressed to the fitting of the two parts (see Figure 4d). Once the glue is cured, we detach the outer molds, and then we use the laser scanner to detect the error between the cured rubber body and the ideal mold (see Figure 4e,f). Finally, we assembled the soft body and a permanent magnet (20 mm × 10 mm × 3 mm) with glue, and a rubber tube was inserted into the end of the actuator and sealed (see Figure 4g,h). Thus far, we have completed the fabrication of the bellows actuator and the worm-like robot with a diameter of 20 mm, length of 31 mm, and weight of 12.5 g, which can realize the steering by adjusting the direction of the magnetic field.

## 3. Force Analysis of the Worm-like Robot

In this section, the motion process of the peristaltic robot is analyzed, and the motion principle of the robot is explained theoretically according to the mechanics theory, as well as the sliding phenomenon during the motion process. Then, the mechanical models of pressure and stretching force, the elongation and stretching force, are established according to the relationship between pressure and elongation obtained in Section 2.

### 3.1. Motion Process Analysis

The motion process of the peristaltic robot is periodic. The robot's motion process can be divided into two stages: inflation and deflation, and can also be divided into two motion modes: extension and contraction, according to the movement state. The steering and guidance of the robot can be realized by adjusting the magnetic field position. We consider the large deformation and nonlinear characteristics of the robot's motion process and the multi-force-field coupling with the magnetic field force and the ground contact friction. It is essential to accurately describe the mechanical principles of the robot's motion process, establish its mechanical model, and control its motion. Ideally, the bellows actuator is subjected to two forces. One can divide into the trust force $F_p$ and radial force $F_e$ caused by pressure, and the other is the contraction force $F_k$ generated by elastic potential energy. According to the robot's motion state, we analyzed the robot's force state and motion phenomenon during the extension, contraction, and steering processes. The robot's motion state and force direction are shown in Figure 5.

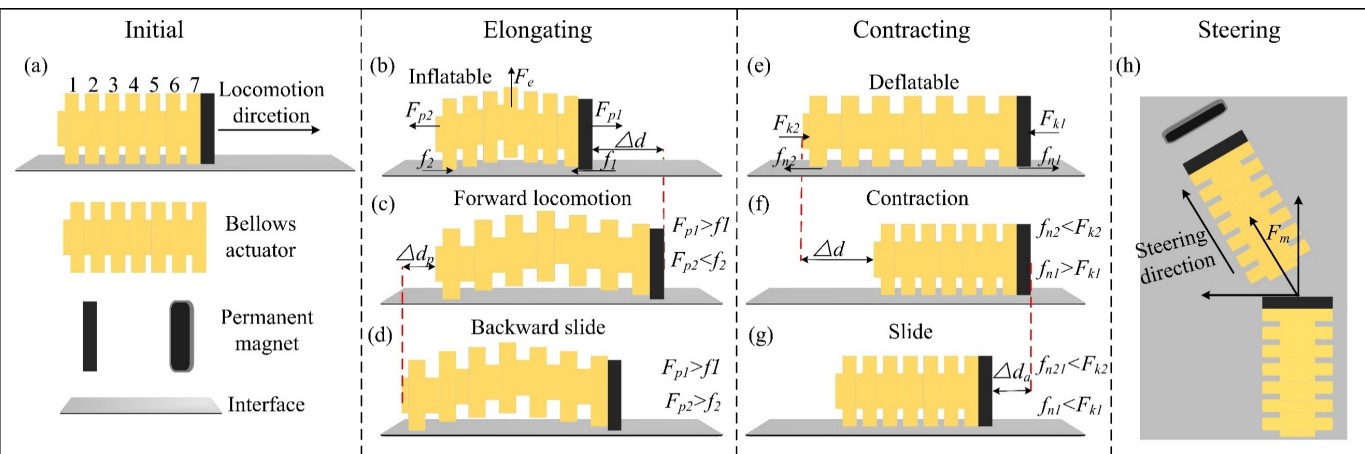

**Figure 5.** The mechanical theory and motion state of the worm-like robot (**a**) Initial state of the robot; (**b**) Initial inflation with without motion; (**c**) Forward movement; (**d**) Backward extension; (**e**) Deflation; (**f**) Forward contraction; (**g**) Sliding movement; (**h**) Magnetic field steering.

As shown in Figure 5a, the bellows driver is divided into seven sections, and the head is equipped with a permanent magnet. Where $F_e$ represents the radial force, $F_{p1}$ and $F_{p2}$ represent the trust force, respectively, which are equal in magnitude and opposite in direction, that is $F_{p1} = F_{p2} = F_P$. $f_1$ and $f_2$ are the friction between the bellows actuator's first section and the motion surfaces, and that between the permanent magnet and the motion surfaces, respectively. $F_{k1}$ and $F_{k2}$ represent the contraction force, respectively, which are equal in magnitude and opposite in direction, that is $F_{k1} = F_{k2}$. $f_{n1}$ and $f_{n2}$ are the total friction force between the seventh section and permanent magnet with the surface, and that between the actuator's first section to the sixth section and the surface, respectively. $F_m$ is a magnetic force, and $\Delta d$, $\Delta d_p$, and $\Delta d_a$ are the theoretical forward step, backward elongation, and slip length of the robot, respectively.

During the initial inflation phase, the robot does not move due to $F_{p1} < f_1$ and $F_{p2} < f_2$. While the bellows bend upwards and deform as a result of the radial force, only the bellows first section and the permanent magnet are in contact with the crawling surface (see Figure 5b). Owing to the permanent magnet and the silicone having different friction coefficients with the crawling surface, the maximum static friction between the silicone and the crawling surface is much larger than the maximum static friction between the permanent magnet and the surface, namely $f_1 < f_2$. With the increase in pressure, the thrusts $F_{p1}$ and $F_{p2}$ are gradually increased. When $F_{p1} > f_1$ and $F_{p2} < f_2$, the robot moves forward, its forward extension is recorded as the theoretical step length of the robot movement $\Delta d$ (see Figure 5c). If we continue to inflate and increase the pressure at this time, when $F_{p1} > f_1$ and $F_{p2} > f_2$, the bellows is stretched in both directions at the same

time. The backward elongation is recorded as $\Delta d_p$ (see Figure 5d), and the robot will be difficult to control at this point. Thus, this situation should be avoided in actual motion.

Nevertheless, when the actuator deflated, with the bellows thrust force and the radial force disappeared, each segment of the bellows came in contact with the crawling surface and contracted with the fourth section of the bellows as the center under the contraction force $F_{k1}$ and $F_{k2}$ (see Figure 5e). Ideally, when $F_{k1} < f_{n1}$ and $F_{k2} > f_{n2}$, the robot contracted along the forward direction, the contraction length is consistent with the elongation length (see Figure 5f). However, in actual motion, slippage often occurred as a result of $F_{k1} > f_{n1}$ and $F_{k2} > f_{n2}$. In this case, both ends of the robot contracted to the center simultaneously, and the shrinkage of the head towards the center is considered as the sliding length of the robot (see Figure 5g). The slippage significantly reduces the crawling efficiency of the robot, which it should avoid as much as possible in the actual movement. The steering movement of the robot mainly relies on the guidance of the magnetic field force. As shown in Figure 5h, a permanent magnetic field is applied near the robot's motion path. Since the robot head contains a permanent magnet, under the magnetic field force $F_m$, the head magnet generates a torque to drive the robot's motion path changed along with the direction of the magnetic field, which can achieve the navigation purpose proposed in this paper.

### 3.2. Mechanical Model

From the force analysis of the motion process, it can be seen that the thrust force $F_p$ and contraction forces $F_k$, and the magnetic field force $F_m$ play a decisive role in the elongation and contraction of the bellows during the robot's motion process. The thrust force can be written as:

$$F_p = PS \tag{10}$$

where $P$ is the pressure, $S$ represents the force-bearing area in the bellows chamber. The contraction force can be calculated by Hooke's law, and it can be written as:

$$F_k = k\Delta l = k(l_a - l) \tag{11}$$

where $k$ represents the spring coefficient.

According to Equations (3), (5), and (9), the relationship between the force-bearing area of the bellows and the pressure can be obtained as:

$$S = \pi r_a{}^2 = \pi \left(\frac{rl}{l_a}\right)^2 = f_s(f^{-1}(p)) \tag{12}$$

Then the mathematical model between the thrust force and the pressure can be written as follows:

$$F_p = P f_s(f^{-1}(p)) \tag{13}$$

Ignoring the friction force, the relationship between the contraction force and pressure can be obtained by the equilibrium of forces as follows:

$$F_k = F_p = P f_s(f^{-1}(p)) \tag{14}$$

Assuming the magnets have homogenous magnetization, without eddy current, and the relative permeability is unity. Magnets placed in the magnetic field space will be subjected to magnetic force and magnetic torque [22,23]. The calculation formula of magnetic force and magnetic torque can be expressed as:

$$\vec{F} = (\vec{m} \cdot \nabla)\vec{B} \tag{15}$$

$$\vec{T} = \vec{m} \times \vec{B} \tag{16}$$

where $m$ is the magnetic moment of the magnetic dipole, $B$ represents the magnetic induction intensity generated by the permanent magnet, and $\nabla$ is used to represent the magnetic induction intensity gradient.

By applying an external magnetic field to the worm-like robot, the head magnets and the external magnetic field will attract each other under the action of magnetic torque to drive the whole robot to change its motion direction, which can achieve the purpose of steering and navigation of the robot. According to Equations (15) and (16), the formula for calculating the magnetic torque can be obtained as:

$$F_m = MBV \sin \alpha \tag{17}$$

where $M$ represents the magnetization of the magnetic material, $V$ is the volume of the magnet, and $\alpha$ represents the angle between the direction of magnetization and the direction of the external magnetic field.

## 4. Results and Discussion

The bellows' performance is tested and compared with the simulation results in this section. Following that, the crawling performance of the robot is tested on different rough surfaces and different environments, and the force status of the robot in the motion state is verified through the experimental results. Finally, we test the navigation function of the robot in the magnetic field environment.

### 4.1. Characteristics of Bellows

The bellows actuator is the only driving device of the robot, which provides the main output force for the robot's movement, and its performance determines the movement performance of the robot. To perform the elongation, expansion, and thrust force with the bellows actuator, we sample multiple dates by inflating the bellows actuator to air pressure, ranging between 0 and 50 kpa, in intervals of 5 kpa, resulting in 30 pressure combinations. We repeat this procedure three times per type and take the average value as its actual value to compare with the simulation results. The results are shown in Figure 6 and Table 2.

As shown in Figure 6 and Table 2, the trend variation of the actuator's actual elongation with air pressure is almost consistent with the simulation (Figure 6a). In the low-pressure phase, the elastic potential energy overcome by the thrust force is smaller, and the rate of change of the elongation is more significant. However, the elastic potential energy overcome is more extensive as the air pressure increases. Thus, the rate of change of the expansion is reduced compared with the initial stage. Moreover, with the increase of air pressure, the error between the actual elongation and the simulation data gradually decreases, and the error rate at 50 kpa is only 1.5%. However, due to the manufacturing error, as shown in Figure 6d, the cross-sectional error is up to 0.3 mm, equivalent to the increase in its thickness $t$. Therefore, the actual expansion is smaller than the simulation, and the expansion error of the driver gradually increases at the low-pressure stage. But when the pressure exceeds 30 kpa, the expansion error decreases. This is because its radial force gradually increases with the pressure, but the error rate still reaches 28.7% at 50 kpa. No matter what, while ensuring the actuator elongation, the expansion is reduced to a certain extent, which positively correlates with the robot's motion performance. Assuming that the bellows do not generate radial deformation, that is, $S$ is a constant value, it can be obtained from Equation (10) that the air pressure with a linear change in the thrust force. In this experiment, the bellows are placed in an acrylic pipeline with an internal diameter of 21 mm to limit its radial deformation. The bellows thrust force measured by the force sensor is shown in Figure 6c. With the increase of pressure, the error between the driver thrust force and the theoretical value increases gradually, but the error tends to be stable after 30 kpa and reaches 1.39N at 50 kpa. Besides the theoretical model's error, the radial deformation and the friction inside the pipeline are also responsible for the thrust reduction.

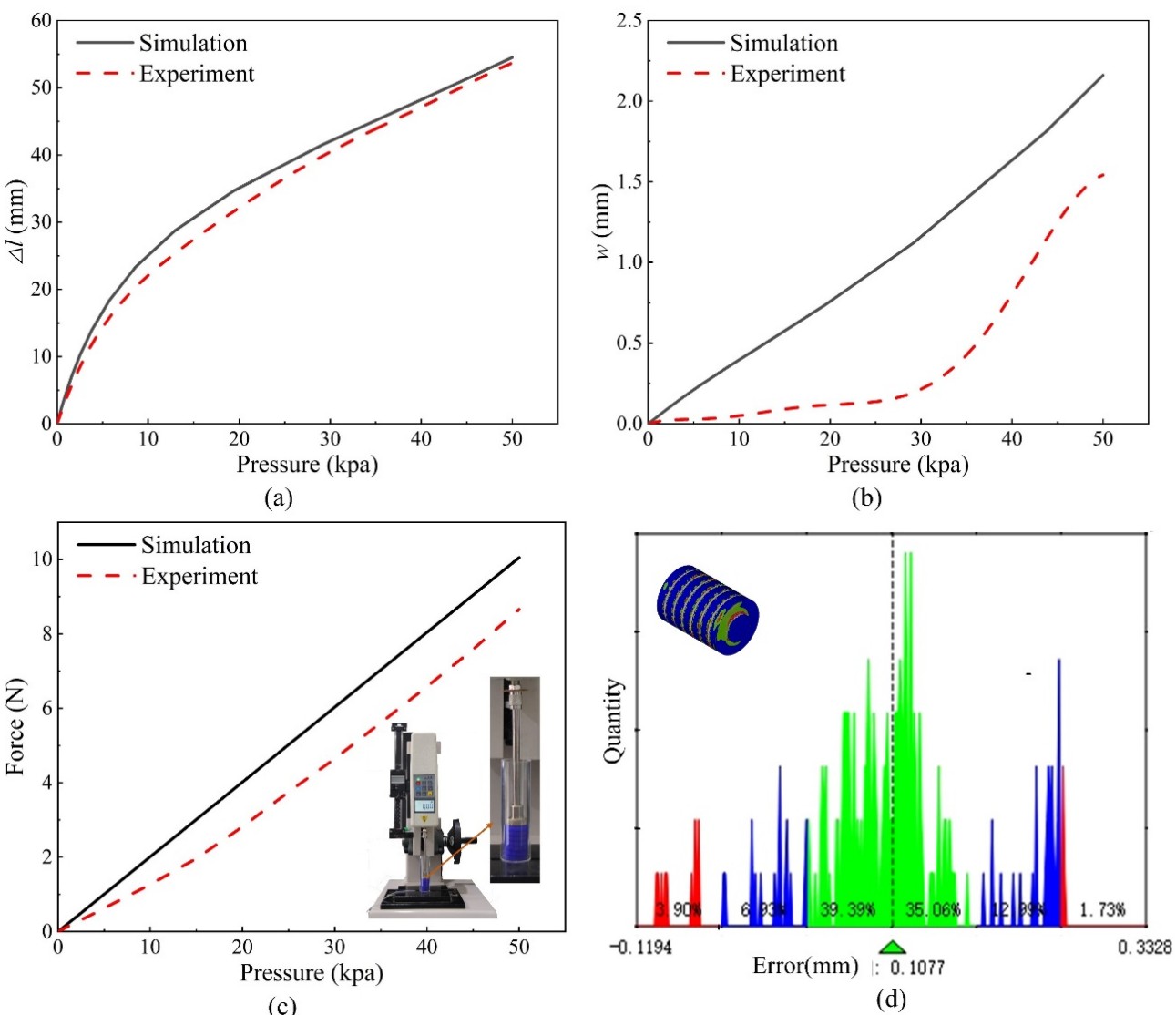

**Figure 6.** The bellows actuator's actual performances (**a**) The actuator's actual elongation performance; (**b**) The actuator's actual expansion performance; (**c**) The actuator's actual thrust force performance; (**d**) The histogram of the actuator's robot section manufacturing error.

**Table 2.** Comparison of the bellows experiment and simulation data.

| Characteristics<br><br>Parameters | Max Elongation (mm) | Max Expansion (mm) | Max Force (N) |
|---|---|---|---|
| Simulation | 54.5 | 2.16 | 10.05 |
| Experiment | 53.68 | 1.54 | 8.66 |

### 4.2. Characteristics of Movement

In order to demonstrate the different motion states caused by different forces during the motion process in Figure 5, we test the single-cycle motion state of the peristaltic robot on different surfaces, including the smooth cardboard and smooth acrylic pipelines, by inflating the bellows actuator to air pressure, ranging between 0 and 50 kpa, in intervals of 5 kpa, respectively. The motion efficiency under different air pressures is calculated by $\eta = \frac{(\Delta d - \Delta d_a)}{\Delta d}$ to evaluate the robot's crawling performance [24]. The results are shown in Videos S1 and S2, Figure 7, and Table 3.

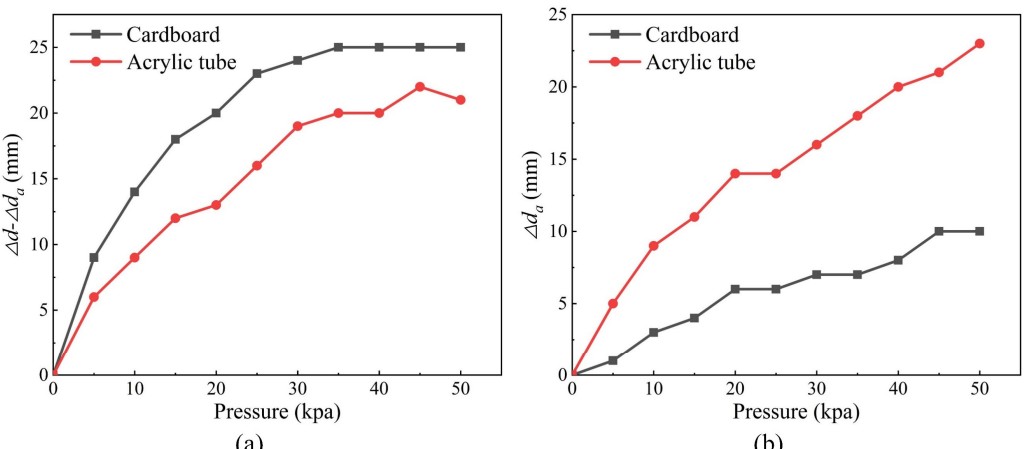

**Figure 7.** The robot's motion performance with different surfaces (**a**) The real step length with different surfaces; (**b**) The sliding length with different surfaces.

**Table 3.** The robot's motion efficiency with different surfaces.

| Pressure (kpa) | 5 | 10 | 15 | 20 | 25 | 30 | 35 | 40 | 45 | 50 | Average |
|---|---|---|---|---|---|---|---|---|---|---|---|
| Cardboard | 0.9 | 0.83 | 0.82 | 0.77 | 0.79 | 0.77 | 0.78 | 0.76 | 0.71 | 0.71 | 0.78 |
| Acrylic tube | 0.55 | 0.5 | 0.52 | 0.48 | 0.53 | 0.54 | 0.53 | 0.5 | 0.51 | 0.48 | 0.51 |

It can be seen from Video S1 that when moving on the smooth cardboard, the robot stretches forward without apparent backward sliding phenomenon as the results in $F_{p1} > f_1$ and $F_{p2} < f_2$, which is consistent with the motion state in Figure 5c. As the air pressure increased, the thrust force increased accordingly. When the air pressure exceeds 25 kpa, the robot shows an apparent backward sliding, and the sliding length increases gradually with the increase of pressure, which is consistent with the motion state in Figure 5d. Since the friction coefficient between the silicone rubber and the acrylic plate is greater than that between the silicone rubber and the smooth cardboard, that is $f_{y1} > f_{z1}$ ($f_{y1}$ and $f_{z1}$ are the friction force between the silicone rubber and the acrylic plate, and the friction force between the silicone rubber and the smooth cardboard, respectively), the pressure of the robot to slip backward is larger than that of the cardboard when moving on the smooth acrylic pipelines. As shown in Video S2, the robot is still without an evident backward slip phenomenon on the acrylic pipeline when the air pressure is inflated to 30 kpa, which thoroughly verifies the correctness of our force analysis during the robot movement.

According to Equations (10), (13), and (14), it can be deduced that with the growth of air pressure, the bellows contraction force will gradually increase as the thrust force increases. Thus, when $f_{n1}$ and $f_{n2}$ are constant, the slip length $\Delta d_a$ will also increase as the air pressure increased, and the motion efficiency gradually decreases. Furthermore, the slip length of the robot in the acrylic pipeline is larger than that on the cardboard because the friction coefficient between the head permanent magnet and cardboard is greater than that between the head permanent magnet and acrylic pipeline, namely $f_{yn1} < f_{zn1}$ ($f_{yn1}$ and $f_{zn1}$ represent the total friction force between the seventh section and permanent magnet with acrylic pipe and smooth cardboard, respectively.). As shown in Figure 7 and Table 3, with the increase of air pressure, the actual step size of the robot in the two environments increases while the slippage gradually increased, and the slip length on the acrylic pipeline is much more considerable than that on the smooth cardboard. Moreover, with the growth of slippage, the motion efficiency of the robot gradually decreases and stabilizes in an interval, which fully proves the correctness of the mechanical model.

In order to prevent the sizeable backward movement of the robot during the movement process, which may cause instability of the movement, we select 25 kpa as the driving pressure of the robot through Figure 7 and Table 2 on the basis of ensuring the movement

efficiency and adequate step size. The frequency of the driver inflating and deflating is 2 s per cycle, which was applied to evaluate the movement speed of the robot under the premise of fully considering the length of the tube and the efficiency of inflating and deflating. The results are shown in Figure 8, and the motion process is shown in Video S3. According to the equation $speed = \frac{dis\tan ce}{times}$, it can be calculated that the crawling speed of the robot in the smooth cardboard is 9.6 mm/s, and the speed in the acrylic pipeline is 5.1 mm/s.

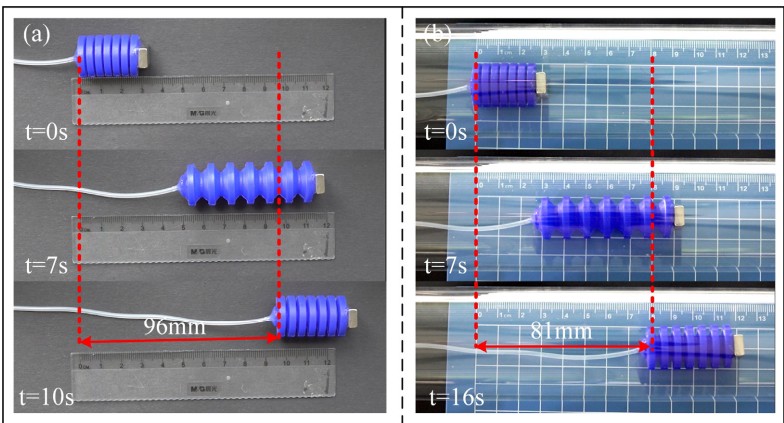

**Figure 8.** The robot's motion speed with different surfaces (**a**)The robot's motion speed on cardboard; (**b**) The robot's motion speed on cardboard acrylic tube.

To further demonstrate the robot's motion performance, we put the peristaltic robot into acrylic pipelines with inner diameters of 21 mm, 32 mm, 40 mm, and 50 mm for linear motion, and the slope motion was carried out in the pipes with a slope of 6°. The results are shown in Figure 9, and the motion process is shown in Video S4. Compared with the robot relying on expansion and contraction, the robot designed in this study has broader applicability. It can move on a particular slope, showing superior motion performance and environmental adaptability.

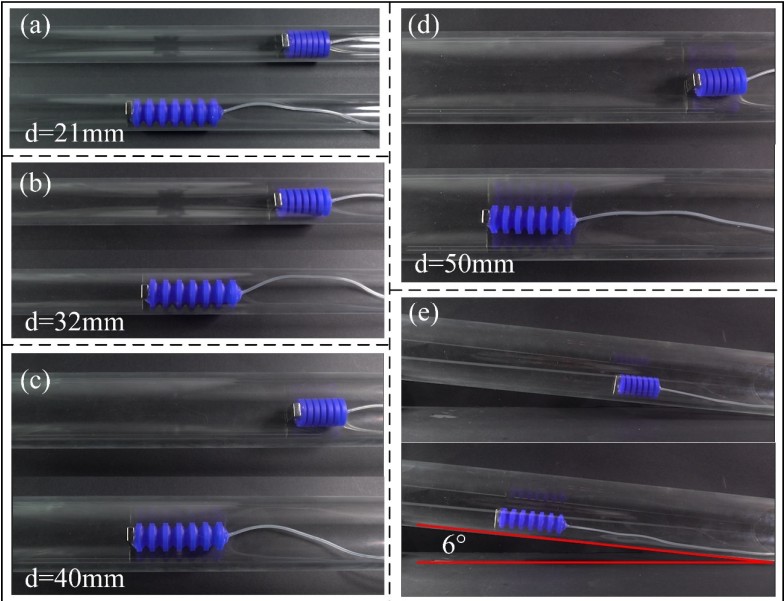

**Figure 9.** The crawling performance of pipelines with different diameters and slope (**a**) The inner diameter of 21 mm; (**b**) The inner diameter of 32 mm; (**c**) The inner diameter of 40 mm; (**d**) The inner diameter of 50 mm; (**e**) The slope angle with 6°.

### 4.3. Characteristics of Steering

Compared with conventional peristaltic robots for steering by adopting a complex multi-module and multi-chamber structure design and controller, inspired by the use of geomagnetic field navigation during the migration of migratory birds, this study utilizes an external magnetic field to guide and turn the robot and realize the navigation function. As shown in Figure 10, a permanent magnet (19 mm × 19 mm × 19 mm, a weight of 48 g) was placed on both sides and in front of the robot's motion path, and driving air pressure was applied to test the robot steering and navigation functions. The results are shown in Figure 10a–c and Video S5. When the magnetic field is the same as the initial direction, the robot moves linearly along with the initial motion direction. Then, we adjust the position of the magnetic field. The permanent magnet on the head of the robot receives the force of the magnetic field. It generates a torque along the direction of the magnetic field, which makes the robot rotate in the direction of the magnetic field and eventually along the magnetic field direction, which shows the effectiveness of magnetic field navigation.

To further verify the performance of the magnetic field navigation proposed in this paper, the robot was placed in a Y-shaped pipeline with an angle between the two straight pipelines. Its motion states with a single magnetic field and dual magnetic fields were tested, respectively. As shown in Figure 10d–f and Video S6, when a magnetic field is added to the side of the Y-shaped pipeline, the robot can realize the directional movement in the pipeline along the magnetic field direction under the traction of the magnetic field force. Furthermore, when the different intensities of the magnetic field were applied on both sides at the same distance, the robot could move in the direction of a stronger magnetic field according to the strength of the magnetic field. Compared with using multiple complex structures and controls to realize robots turning, the magnetic field navigation method proposed in this paper is more straightforward in design and stronger in environmental adaptability.

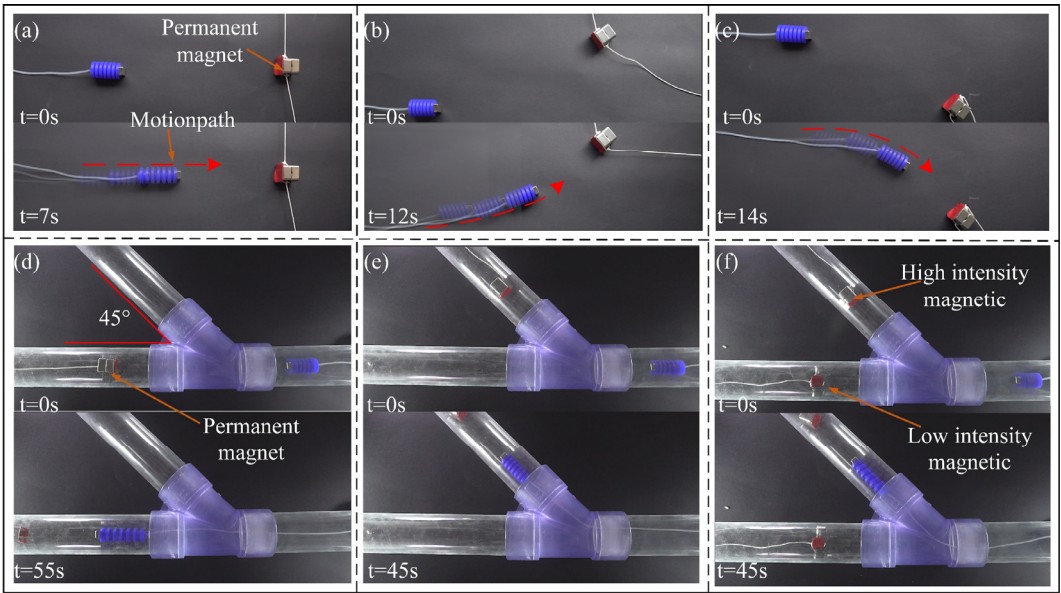

**Figure 10.** The magnetic navigation performance in different environments (**a**) Crawl in a straight line on the cardboard; (**b**) Turn left on the cardboard; (**c**) Turn right on the cardboard; (**d**) Crawl in a straight line on the pipeline; (**e**) Turn right on the pipeline; (**f**) Steering along the direction of the strong magnetic field under different magnetic field strengths.

## 5. Conclusions

In this work, inspired by the migration of migratory birds that can use the geomagnetic field for navigating, we presented a single-joint centimeter-scale peristaltic robot that can be navigated by adjusting the position of magnetic fields. The proposed robot was

driven by air pressure, and the bellows were selected as the driver structure. Through the Yeoh constitutive model, we established the relationship between the elongation of the bellows and the air pressure, and the finite element software was used to analyze the influence of the chamber's number and thickness on the bellows' performance, which provides a basis for the selection of the bellows structural parameters. Moreover, the force state of the peristaltic robot under the motion states of elongation, contraction, and rotation was analyzed using the mechanics theory, and the models of air pressure and thrust force, air pressure, and contraction forces were established, providing supporting theories for experimental phenomena. Finally, through experiments, the actual bellows performance, robot motion performance, and magnetic field navigation performance have been verified, respectively, and the results are in complete agreement with the theoretical analysis. In addition, the experimental results also show the robot's superior environmental adaptability, which can move in acrylic pipelines with inner diameters of 21 mm, 32 mm, 40 mm, and 50 mm, and can make slope movements in a pipeline with a slope of $6°$. As well as moving on surfaces with different roughness, its maximum speed can be 9.6 mm/s. It can also effectively realize the direction selection in planes and complex pipelines according to the magnetic field. These results effectively prove the robot's superiority in the application and structure design scope and show great potential for pipeline detection applications.

Although the robot showed excellent performance, there was still room for improvement. For example, the robot can only move on a two-dimensional plane and cannot crawl vertically because of its single-joint structure and the need to rely on friction to move. The magnetic field of the permanent magnet is difficult to control and realize direction guidance accurately, and the open loop control cannot obtain the actual motion state. In the future, we will further expand the robot's functions and use electromagnetic fields integrated with sensor feedback to achieve precise navigation and control of the robot. Some practical application experiments, such as pipeline detection and ruins search and rescue, will also be realized in future work.

**Supplementary Materials:** The following supporting information can be downloaded at: https://www.mdpi.com/article/10.3390/machines10111040/s1, Video S1: The single-cycle motion state of the peristaltic robot on cardboard; Video S2: The single-cycle motion state of the peristaltic robot on the acrylic pipeline; Video S3: The robot's motion speed with a different surface; Video S4: The crawling performance of pipelines with different diameters and slope; Video S5: The magnetic navigation performance in cardboard; Video S6: The magnetic navigation performance in the pipeline.

**Author Contributions:** Conceptualization, D.M.; methodology, D.M.; software, J.W.; validation, D.M. and X.Z.; investigation, G.T., C.Z. and D.M.; data curation, D.M., C.L. and X.Z.; writing—original draft preparation, D.M.; writing—review and editing, D.M. and Y.W.; visualization, X.Z. and G.T.; supervision, Y.W.; project administration, Y.W. All authors have read and agreed to the published version of the manuscript.

**Funding:** This research was supported by the National Key Research and Development Program of China (2020YFB1312900), the National Natural Science Foundation of China (51975184), the Changzhou Sci & Tech Program (CE20215051), and the Fundamental Research Funds for the Central Universities (B210202124). The authors gratefully acknowledge the support.

**Data Availability Statement:** Not applicable.

**Acknowledgments:** The authors are grateful to the editor and reviewers for their constructive comments and suggestions, which have improved this paper.

**Conflicts of Interest:** The authors declare no conflict of interest.

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
