# Peer review of "A Single-Joint Worm-like Robot Inspired by Geomagnetic Navigation"

_machines, doi:10.3390/machines10111040_

Round 1

Reviewer 1 Report

In order to improve the paper, the reviewer asks the authors to specify:

- the weight of the mobile robot, 

- if it is possible for the material used to age over time and influence the performance of the robot

Corrections:

-in table 1, Heignt (l. mm), Raaidus (r. mm) must be corrected with Height (l. mm) Radius (r. mm);

-in Fig. 5 b) the term with not motion should be replaced with without motion ;

-in Fig. 10 the term should be crawl instead of craw

Reviewer 2 Report

The geomagnetic field inspired the authors to design a soft robot that can be easily guided by a magnet. The author showed the worm-like robot move on the cardboard with a speed of 9.6 mm/s and is able to move on different rough surfaces and a titled pipeline. However, from the evaluation of the mechanical design and results, I don’t find enough novelty for this paper to publish. Some of the concerns are listed below:

1.     The robotic design is not novel. The cylindrical extension soft robotic design has been published for other purposes. For example:

Zhang, Boyu, et al. "Worm-like soft robot for complicated tubular environments." Soft robotics 6.3 (2019): 399-413.

Applying walking or extending in the tubular environment is not new either.

2.     The magnetic guiding method looks interesting. However, it has an intrinsic drawback: the robot can’t go in our desired direction by itself. Therefore, the robot requires a magnet to guide, and an air tube at the back to serve as a driven method.

3.     Theoretically, the robot can go forward without pneumatic driven because the magnet can make it move in the direction automatically, it could be much faster than the driven method.

4.     The robot is demonstrated to move on the paper board and rough surface, while the robotic design didn’t show any clue about holding the robot on the surface, which means, the robot is moving based on friction.

5.     Since the robot is moving because of friction, the robot won’t do well on a titled pipeline, I would expect the robot will fall if there is a larger hill, the friction won’t help to move upwards.

Round 2

Reviewer 2 Report

The current modified form looks good. I will still keep the comment for the first review, the drawback of the design and application can be improved in their future work.